# DynaDojo: An Extensible Benchmarking Platform for Scalable Dynamical System Identification

**Logan Mondal Bhamidipaty**[*]
Stanford University
loganmb@cs.stanford.edu

**Tommy Bruzzese**[*]
Stanford University
tbru@cs.stanford.edu

**Caryn Tran**[*]
Northwestern University
caryn@u.northwestern.edu

**Rami Mrad**
UC Berkeley
ramiratlmrad@berkeley.edu

**Max Kanwal**[†]
Stanford University
kanwal@stanford.edu

## Abstract

Modeling complex dynamical systems poses significant challenges, with traditional methods struggling to work across a variety of systems and scale to high-dimensional dynamics. In response, we present DynaDojo, a novel benchmarking platform designed for data-driven dynamical system identification. DynaDojo enables comprehensive evaluation of how an algorithm's performance scales across three key dimensions: (1) the number of training samples provided, (2) the complexity of the dynamical system being modeled, and (3) the training samples required to achieve a target error threshold. Furthermore, DynaDojo enables studying out-of-distribution generalization (by providing multiple test conditions for each system) and active learning (by supporting closed-loop control). Through its user-friendly and easily extensible API, DynaDojo accommodates a wide range of user-defined `Algorithms`, `Systems`, and `Challenges` (scaling metrics). The platform also prioritizes resource-efficient training for running on a cluster. To showcase its utility, in DynaDojo `0.9`, we include implementations of 7 baseline algorithms and 20 dynamical systems, along with many demo notebooks. This work aspires to make DynaDojo a unifying benchmarking platform for system identification, paralleling the role of OpenAI's Gym in reinforcement learning.[1]

## 1 Introduction

Dynamical systems, fundamental to disciplines like physics, engineering, economics, and neuroscience, are difficult to predict and control when nonlinear and high-dimensional. Traditional methods that rely on a known underlying model structure fall short when faced with modern problems like stock market forecasting or modeling human social interactions, where the structure is either unknown or non-existent. This has prompted a shift toward data-driven modeling (*system identification*), and especially model-free methods, bypassing the need for predefined equations [1]. To benefit from these data-driven approaches, however, researchers and practitioners need tailored benchmarks to easily evaluate and compare methods for system identification in their area of study.

In this work, we present DynaDojo, an open, extensible benchmarking platform to standardize the process of benchmarking *any* learning algorithm on *any* dynamical system. Modeled after OpenAI's Gym [2] and Procgen [3], DynaDojo introduces abstractions over algorithms, systems, and challenges

---

[*]equal contribution

[†]corresponding author

[1]https://github.com/DynaDojo/dynadojo

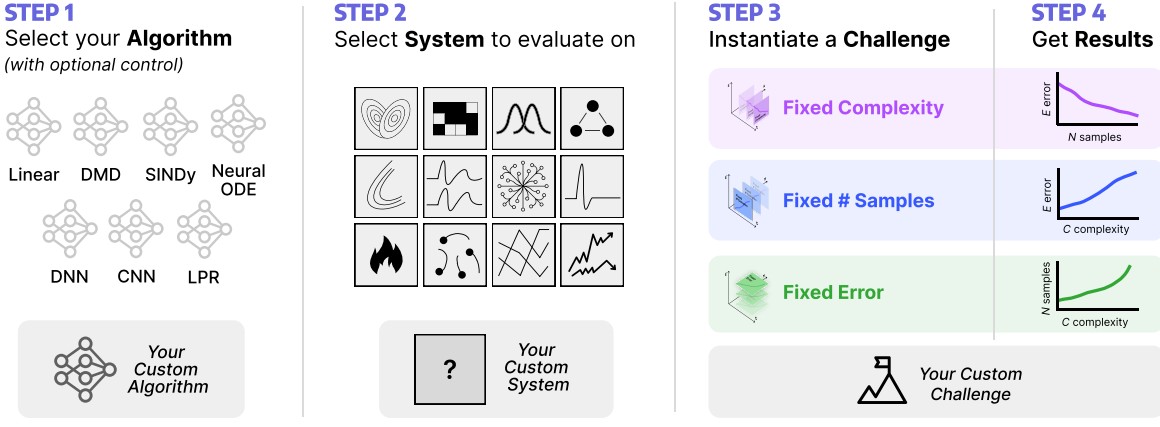

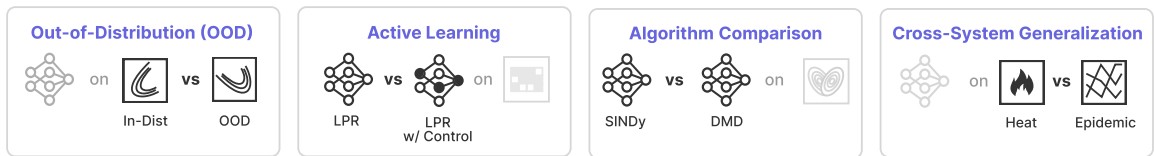

Figure 1: Pipeline for how to use DynaDojo. Select an algorithm (Step 1) and a system (Step 2), then instantiate a challenge (Step 3), and evaluate to get results (Step 4). It is easy to run repeated DynaDojo challenges to compare performances on in- vs. out-of-distribution data, with vs. without active learning, between algorithms, and across systems (Step 5).

to promote ease of use, modularity, and broad compatibility. It includes a growing library of reference algorithms and tunable dynamical systems. Uniquely, DynaDojo facilitates benchmarking scalability via procedural generation and customizable challenges that systematically vary training set size, system complexity, and target error rates. Challenges also allow for assessing out-of-distribution generalization, active learning, algorithm comparison, and cross-system performance. Through its simple yet flexible API, parallelized execution, and interactive demos, DynaDojo aims to serve as an accessible yet rigorous benchmarking platform for the system identification community.

## 2 Related work

Because of the shared contexts in the fields of system identification and reinforcement learning (RL), we draw upon and connect literature from both areas to motivate the development of our benchmarking platform. While system identification focuses on accurately modeling system dynamics, RL aims to optimize inputted actions to a system to maximize reward. Despite their distinct objectives, they both seek to model and interact with complex environments and can often be used to solve similar or overlapping problems. For a summary comparison of DynaDojo against existing benchmarks, see Table 1.

### 2.1 Benchmarks and benchmarking platforms

**Single-System Benchmarks** In the field of system identification, benchmarks have been created that use data on one specific dynamical system from a physical phenomenon of interest [4, 5]. These single-system benchmarks have been used to evaluate specific learning algorithms [6] or to compare different approaches [7]. These narrowly focused benchmarks do not facilitate the evaluation of an algorithm's generalization across a diverse set of systems, which is a central aim of our work.

**Benchmark Suites** Benchmark suites offer a broader scope for evaluating system identification algorithms by covering a larger subset of system classes. These suites, however, are restricted to specific types of systems or representations, such as chaotic systems [8, 9], physical systems [10], or partial differential equations [11, 12]. DynaDojo is agnostic to the type of dynamical system to allow

| | Systems | Algorithms | Extensible | OOD | Active Learning | Sample Efficiency | Complexity Measure |
|---|---|---|---|---|---|---|---|
| Single-System [4, 5] | 1 | | | | | | |
| Chaos [8] | 131 | | | | | | Fixed |
| PDEArena [11] | 5 | 13 | ✓ | | | | |
| PDEBench [12] | 11 | 3 | ✓ | ✓ | | | |
| nn-benchmark [10] | 4 | | ✓ | ✓ | | | |
| Procgen*[3] | 16 | | ✓ | ✓ | ✓ | ✓ | Binary |
| **DynaDojo** | 20 | 7 | ✓ | ✓ | ✓ | ✓ | Tunable |

Table 1: Comparison with related system identification benchmarks. DynaDojo, to the best of our knowledge, is the first extensible dynamical systems benchmark that evaluates out-of-distribution (OOD) trajectories, supports active learning, benchmarks sample efficiency, and implements dynamical systems with tunable complexity measures. *Note: Procgen is an RL benchmark.

for a diverse range of systems to be included. We implement 20 such systems (Figure 2) and provide a simple wrapper interface to use to add more.

**OpenAI Gym**    To address the fragmentation and lack of standardization in system identification benchmarks, we adopt a similar approach to OpenAI's Gym environment [2]. OpenAI Gym offers a common interface for RL benchmarking tasks and, with over 6,500 citations, has become a standard benchmark framework in RL. Our work creates an extensible, standardized gym-like platform to unify system identification benchmarks. Contrary to OpenAI's focus on environments and *not* agents [2], we provide abstractions over both entities (systems and algorithms, respectively) and additionally implement challenges that orchestrate benchmark evaluation while scaling parameters such as system complexity and training set size. In Subsection 2.2 we motivate our focus on scaling system complexity and training samples. And in Section 3.4, we explain why we implement challenges for benchmark orchestration.

## 2.2    Generalization and scaling

In designing DynaDojo, we drew upon a variety of literature to determine desirable features and evaluation metrics to implement in our platform.

**Out-of-Distribution Generalization**    In system identification, there is interest in understanding how well models can generalize beyond the training distribution. [13] probes how deep learning models generalize to trajectories from out-of-distribution initial conditions for dynamical systems. [14] investigates whether deep neural networks learning cellular automata show out-of-distribution generalization for unseen initial configurations with different rule sets. Recent RL benchmarks have sought to split train and test data to draw from different gaming environments in response to problems of overfitting on training environments [15, 16]. Motivated by this work, DynaDojo enables evaluating algorithms on both in-distribution and out-of-distribution data.

**Cross-System Generalization**    While we are unaware of any work in system identification that explores generalization across different systems, the concept has been explored in RL. Generalization in RL has expanded from considering unseen states to learning across different domains, as exemplified by AlphaZero's ability to learn Go, Chess, and Shogi [17], compared to AlphaGo's specialization in Go [18]. It is also similar the field of multi-task learning in which one general policy might train on several different environments [19]. This trend is reflected in RL benchmarks and toolkits that measure performance across a diverse variety of environments [2, 3]. Our work takes an analogous approach by facilitating easy evaluation of an algorithm's performance across different classes of dynamical systems, thereby capturing the algorithm's cross-system generalization capabilities.

**Scaling Complexity**    There is a desire to understand how algorithms perform on systems of varied complexity. For example, within system identification, [14] and [20] test deep neural networks on their generalization capabilities for learning cellular automata with varied neighborhood sizes.

In RL, benchmarks have been designed to train and test algorithms on game environments with varying difficulty levels [3]. This work on games, however, uses imprecise notions of difficulty which

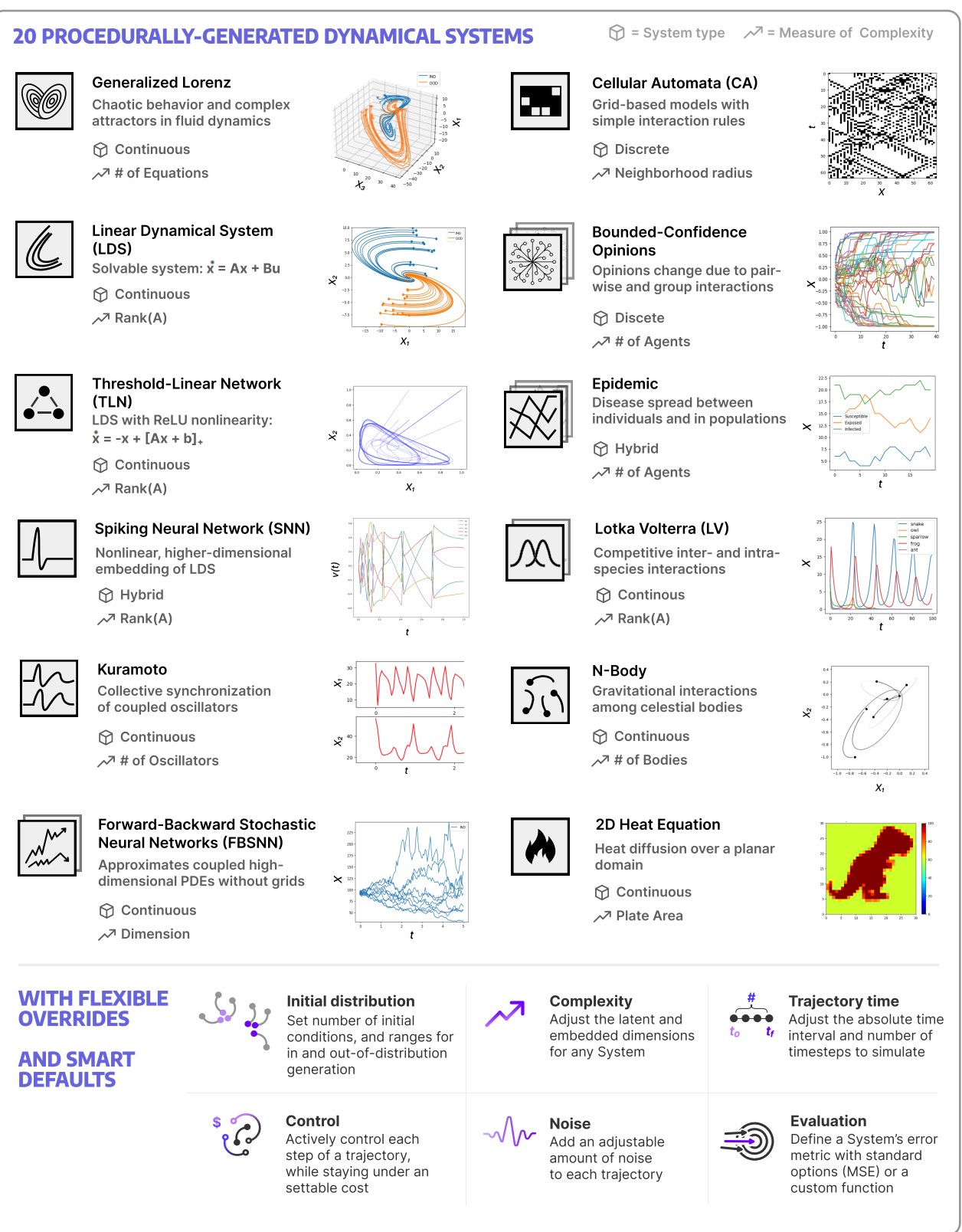

Figure 2: The 20 dynamical systems currently packaged in DynaDojo. Systems are annotated with the system type (discrete, continuous, or hybrid) and their measure of complexity. Users can adjust nearly all DynaDojo parameters on these systems, including how trajectories are initialized, simulated, and controlled, as well as system complexity and evaluation.

are calibrated roughly on the training speeds of baseline algorithms, but this definition is inconvenient because it is hard to generalize and test.

In system identification, [8] provides a suite of over 100 benchmark datasets of known chaotic dynamical systems. Each system is annotated with mathematical properties reflecting the complexity of the system. These annotations facilitate comparison of learning methods across dynamical systems of varying complexity; our work is similarly motivated. In DynaDojo, instead of providing fixed datasets corresponding to different complexity levels, dynamical system classes are defined to allow their complexity levels to be programmatically scaled. This mirrors the approach in [21] which evaluates physics-informed neural networks on a pair of dynamical systems with tunable parameters that control complexity.

**Active Learning**    In RL, active learning—where the algorithm selects control inputs to intelligently explore state-space—enhances sample efficiency. While dynamical systems can accept control inputs and therefore can support active learning, this property is often underutilized in system identification benchmarks that predominantly use static datasets [4, 5, 8, 10, 12]. In our work, we enable algorithms to provide control inputs to dynamical systems, thereby facilitating active learning approaches and interactive data generation.

**Sample Efficiency**    The ability to learn effectively from a limited number of training samples is particularly relevant as it directly influences an algorithm's performance in real-world scenarios where data may be scarce, expensive to obtain, or hard to simulate [22]. In DynaDojo, we've designed metrics that measure how an algorithm's performance scales with changes in system complexity and training dataset size. This feature enables users to assess an algorithm's sample efficiency, particularly as it handles increasing complexity.

## 3    Overview of DynaDojo

DynaDojo operates on three core objects: *Algorithms*, *Systems*, and *Challenges*. Run a Challenge with any given Algorithm and System to evaluate how a learning algorithm's performance scales (Figure 1). DynaDojo currently provides a suite of 7 Algorithms, 20 Systems, and 3 Challenges to be used. Additionally, DynaDojo provides abstract interfaces for Algorithms, Systems, and Challenges that can be extended to support custom implementations (Figure 3).

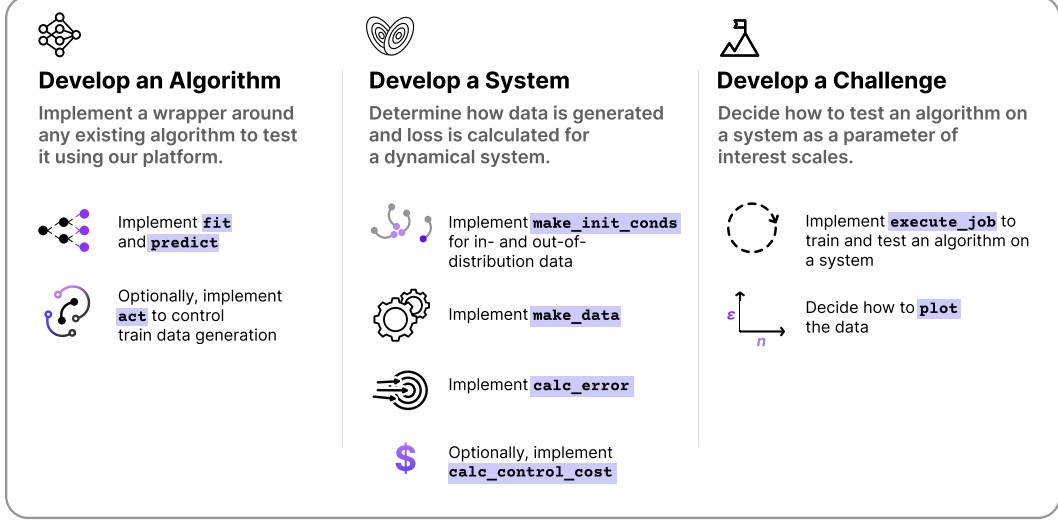

Figure 3: Users can extend DynaDojo by implementing their own Algorithm, System, or Challenge.

### 3.1    Algorithms

*Algorithms* are subclasses of `AbstractAlgorithm`, an interface designed to wrap any learning algorithm that can be used for system identification. At initialization, algorithms are provided the dimensionality of the data which can be used to define the appropriate parameters for the algorithm,

for example, the number of layers in a neural network or degree of a polynomial in a polynomial regression. The `AbstractAlgorithm` interface further abstracts the details of the algorithm under `fit`, `predict`, and `act` methods for ease of use and simplicity. See the online documentation for additional details on the 7 algorithms included in DynaDojo.

## 3.2 Systems

*Systems* are subclasses of `AbstractSystem`, an interface to wrap any procedurally generated dynamical systems. Systems are initialized with a latent dimension and embedding dimension which determine the complexity of the system and its generated trajectories. The `AbstractSystem` interface abstracts the details of simulating the system under the `make_init_conds` and `make_data` methods. Initial conditions can be produced in-distribution or out-of-distribution. Data can be generated with noise or optional control inputs. The `AbstractSystem` interface also abstracts the evaluation metric for any particular dynamical system with the `calc_error` method. For example, continuous systems, such as linear dynamical systems, might implement mean-squared error to evaluate the accuracy of predicted trajectories, whereas a binary system, such as cellular automata, might use Hamming distance instead. Other system-specific metrics, such as achieving stability with control, can likewise be defined in `calc_error`. See Figure 2 for the systems that we package with DynaDojo and the adjustable parameters available for systems.

## 3.3 Challenges

*Challenges* are subclasses of `AbstractChallenge`, an interface to orchestrate the evaluation of algorithms on systems. Challenges simplify and parallelize repeated trials of training and testing algorithms on systems while a parameter (of the system, algorithm, or training process) scales. A challenge is run via the `evaluate` method: This method handles parameter scaling, seed generation (for reproducible execution), and job parallelization. `evaluate` calls on `execute_job`, which is where one defines the protocol for algorithm and system instantiation, data generation, and testing for a single trial. The `plot` method visualizes the results from the challenge evaluation.

In DynaDojo 0.9, we implement three challenges to evaluate scaling: `FixedComplexity`, `FixedTrainSize`, and `FixedError`. In `FixedComplexity`, we repeatedly train and test an algorithm on a system of fixed complexity while scaling the number of training samples. In `FixedTrainSize`, we repeatedly train and test an algorithm on systems of increasing complexity while fixing the number of training samples. In `FixedError`, we search for the number of training samples necessary for an algorithm to achieve a target error rate on systems of increasing complexity. In Figure 4, we visualize the relationship between these three scaling challenges.

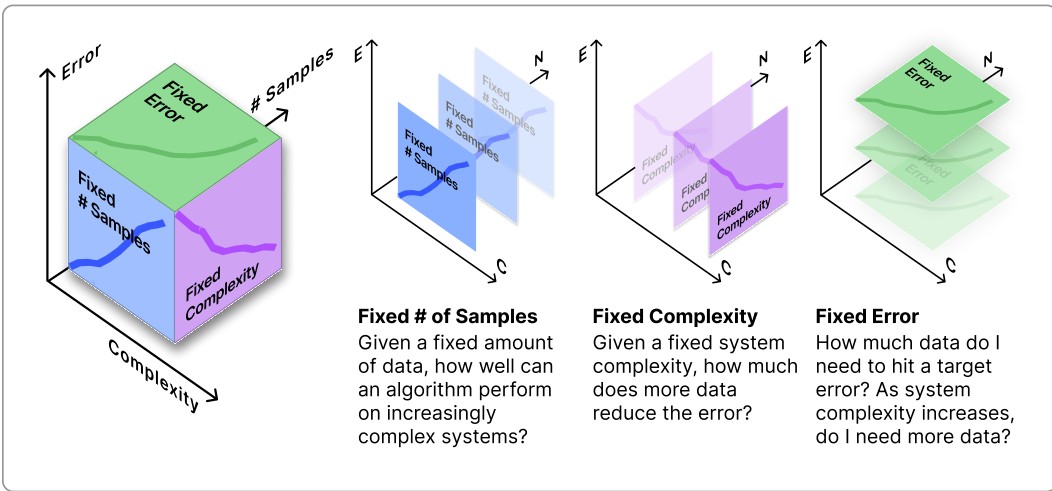

Figure 4: DynaDojo Challenges provide a snapshot of an Algorithm's scaling behavior along one slice of a "performance landscape" relating three dimensions: system complexity (C), prediction error (E), and number of training samples (N). Each Challenge varies parameters along one dimension while holding a second one constant, to measure the algorithm's performance on the third.

## 3.4 Design

**Unified but decoupled.**   Rather than providing a single or suite of benchmarks, DynaDojo is a benchmarking *platform*. We designed abstractions over algorithms *and* systems to unify system identification benchmarking under one platform. This allowed us to package 20 Systems and 7 Algorithms in DynaDojo and standardize comparing them via challenges for better consistency. While each system and algorithm work seamlessly together, they are not locked into the platform. To ameliorate the inflexibility of requiring interfaces over both algorithms and systems, we designed them to be decoupled from another and thus independently usable outside of DynaDojo (see Section 4.1).

**Focused on scaling, not error.**   Often, algorithms are evaluated by their performance on a single task. On the contrary, we are focused on how that performance scales as the task gets harder or more data is provided (see Figure 4). Specifically in system identification, we saw an opportunity to numerically define task difficulty via measures of complexity. To evaluate scaling, many rounds of instantiating algorithms and systems, training, and testing must be repeated as parameters scale. This can lead to embarrassingly parallel resource-intensive workloads. Thus, we conceived of Challenges to manage job parallelization across cores and compute nodes. We also packaged a DynaDojo Docker container for easy use on cluster environments.

**Procedurally generated, not static.**   Rather than a static dataset, dynamical systems in DynaDojo procedurally generate data at train and test time in order to support active learning, where system trajectories are altered by control inputs provided from an algorithm. Additionally, we require a tunable, continuous measure of complexity for each system to support scaling metrics. This requires that dimensionality of system trajectories must be dependent on the system complexity–another reason for procedural data generation. Compared to pre-computed datasets, a limitation of our work is that procedural generation of data can be costly, especially for systems of high complexity.

**Simple by default.**   A key challenge in system identification is being able to compare one algorithm against others across many systems. This is difficult because various existing algorithms and systems either lack a simple API or all have different APIs. Wrestling together APIs and setting the right parameters is a major barrier to benchmarking in system identification. We designed DynaDojo with a focus on simplicity of the API to ensure that different algorithms can painlessly run on any system. To enable this, all systems and algorithms must come with default presets for all parameters, with optional overrides. This requires developers to do work upfront to determine reasonable or adaptive settings for the algorithms or systems they contribute.

**Flexible and versatile.**   We support a variety of settings to enable features such as OOD data generation and control inputs in order to ensure DynaDojo covers broad use-cases (Figure 2). Our extensible interfaces for algorithms, systems, and challenges to ensure that DynaDojo can be adapted to use-cases we have not yet covered (Figure 3). To show the versatility of DynaDojo as a benchmarking platform, we provide numerous example Jupyter notebooks for implemented algorithms, systems, and challenges, available in our GitHub repository.

## 4   Example Usage

### 4.1   Running a single algorithm on a single system

DynaDojo algorithms and systems can be used independently of challenges. To train and test a single algorithm instance on a single system instance, first instantiate the system and create the training and test data (which, in this example, is OOD).

```
1 latent_dim, embed_dim, train_size, test_size, timesteps = (3, 3, 50, 10, 50)
2 lorenz= LorenzSystem(latent_dim, embed_dim, seed=100)
3 x0 = lorenz.make_init_conds(train_size)
4 y0 = lorenz.make_init_conds(test_size, in_dist=False)
5 x = lorenz.make_data(x0, timesteps=timesteps)
6 y = lorenz.make_data(y0, timesteps=timesteps, noisy=True)
```

Then instantiate the algorithm, fit on the training data, predict, and calculate error. Plotting utilities, demos, and examples are provided in our GitHub repository.

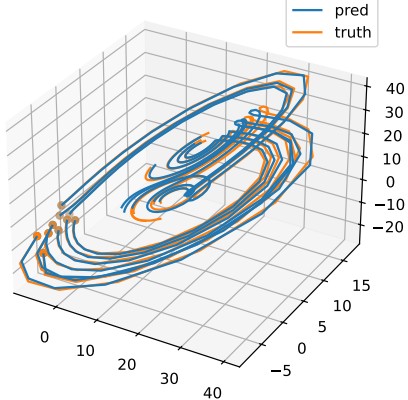

```
1 sindy = SINDy(embed_dim, timesteps,
  ↪   seed=100)
2 sindy.fit(x)
3
4 # predict trajectories
5 y_pred = sindy.predict(y[:, 0],
  ↪   timesteps)
6 error = lorenz.calc_error(y, y_pred)
7
8 fig, ax = dynadojo.utils.plot([y_pred,
  ↪   y], target_dim=min(3, latent_dim),
  ↪   labels=["pred", "truth"])
```

(a) Code for fitting and testing `SINDy` on a `LorenzSystem` using `DynaDojo`. Plotting utilities are provided to visualize high dimensional systems.

(b) Plot of `SINDy` algorithm prediction for a Lorenz system showing high overlap with ground truth.

## 4.2 Running a challenge

To evaluate, for example, how linear regression generalizes to linear dynamical systems of increasing complexity, run `FixedTrainSize`. First, decide on the complexities (latent dimensions) to scale across, the number of training samples, the number of trials, and the training and testing conditions. Supply these arguments to instantiate a `FixedTrainSize` challenge. Then, evaluate the challenge with the algorithm class and plot the results. By default, challenges are run without parallelization; however, code examples showing parallelization across cores and computers are provided on GitHub.

```
1 challenge = FixedTrainSize(
2     Latent_dims=np.logspace(1, 3,
      ↪   20, include_end=True),
3     train_size=100,
4     timesteps=50,
5     control_horizons=0,
6     max_control_cost_per_dim=0,
7     system_cls=LDSSystem,
8     trials=100,
9     test_examples=50,
10    test_timesteps=50 )
11
12 data = challenge.evaluate(
13     LinearRegression,
14     noisy=True,
15     ood=True)
16
17 FixedTrainSize.plot(data)
```

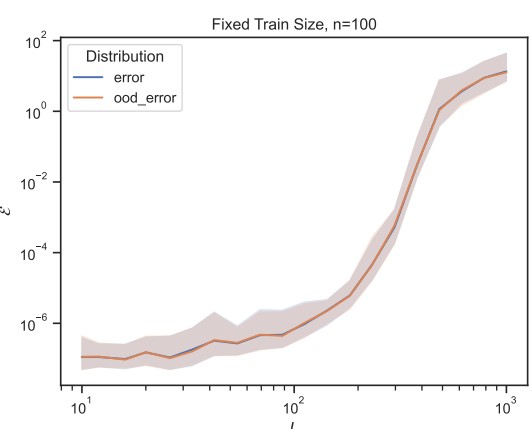

(b) `FixedTrainSize` plot for linear regression on a linear dynamical system with 100 training samples. Latent dimensionality is on the $x$-axis and error on the $y$-axis. Curves for testing on in- and out-of-distribution (OOD) data are overlapping, showing OOD generalization. After L = 100, error rapidly increases, showing weaker sample efficiency with high system complexity.

(a) Code for running `FixedTrain` challenge with on a training set size of 100 for `LinearRegression` on `LDSSystem` with latent dimensions from 10 to 1000.

### 4.2.1 Analyses

With a suite of baseline algorithms and dynamical systems, DynaDojo is designed to support running repeated challenges to analyze algorithms across systems, schematically depicted in Figure 1.

**Out-of-Distribution Generalization**    To test how an algorithm generalizes to out-of-distribution (OOD) data, run a challenge with the `ood` parameter enabled. Challenges will test the algorithm on data simulated *both* from initial conditions drawn from the same distribution—as the training set initial conditions—and from those drawn OOD. See Figure 7 for an example analysis of OOD generalization in a `FixedComplexity` challenge.

**Active Learning**    DynaDojo algorithms can optionally implement an `act` method which generates control inputs that DynaDojo systems accept when generating data. To compare the effect of active learning, run a challenge for a given algorithm without control and the same algorithm with control on a given system. See Figure 8 for an example analysis of active learning in a `FixedError` challenge.

**Comparing Algorithms**    To compare algorithms, run a challenge for two different algorithms on the same system. See Figure 9 for an example comparison of two algorithms in a `FixedError` challenge.

**Cross-System Generalization**    To investigate whether an algorithm's performance is generalizable across systems, run a challenge with an algorithm on two different systems. See Figure 10 for an example analysis of cross-system generalization in a `FixedTrainSize` challenge.

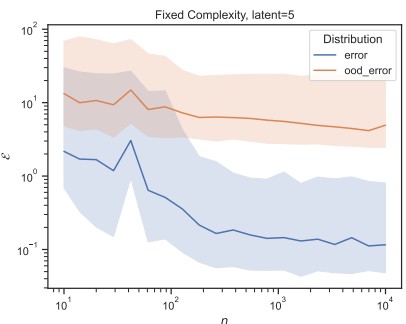

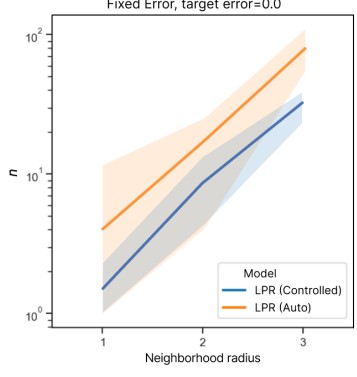

Figure 7: Comparing OOD generalization for deep neural networks (DNN) on linear dynamical systems (latent dimension 5) in a `FixedComplexity` challenge. In-distribution test error (blue line) is decreasing steeply but OOD test error (orange line) is constant as number of training samples ($x$-axis) increases, showing lack of OOD generalization as training size scales.

Figure 8: Comparing active learning for Lowest Possible Radius (LPR) on cellular automata (CA) in a `FixedError` challenge. As CA complexity ($x$-axis) scales, LPR w/ control (blue line) requires less training samples ($y$-axis) than LPR w/o control (orange line) to achieve zero error.

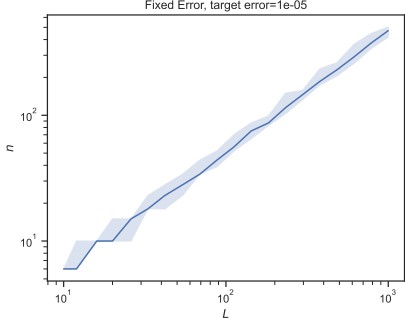

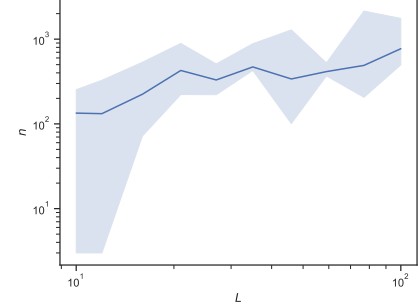

(a) `FixedError` for LR on LDS, target error=1e-5.    (b) `FixedError` for DNN on LDS, target error=1e0.

Figure 9: Comparing two algorithms: Linear regression (LR) and deep neural network (DNN) are evaluated on linear dynamical systems in a `FixedError` challenge. As complexity ($x$-axis) of the system scales, LR (left) achieves a much lower target error of $10^{-5}$ with fewer or comparable number of training samples ($y$-axis) than needed by a DNN (right) for a *higher* target error rate of $1$.

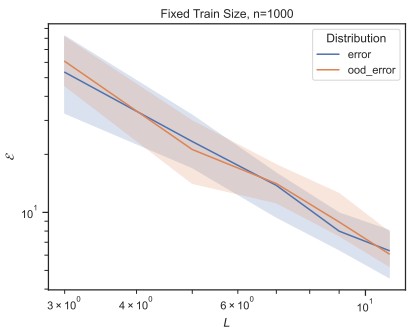
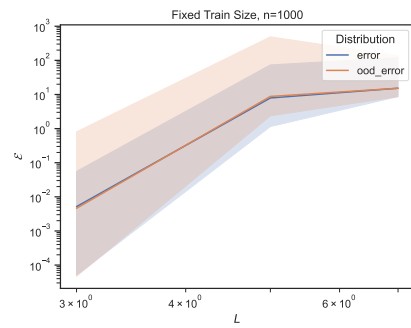

(a) `FixedTrainSize` with SINDy and Lorenz.  (b) `FixedTrainSize` with SINDy and LDS.

Figure 10: Cross-system generalization: SINDy is evaluated on Lorenz Systems (left) and linear dynamical systems (LDS) (right) in a `FixedTrainSize` challenge. Error ($y$-axis) decreases with complexity ($x$-axis) for Lorenz, but increases and plateaus with complexity for LDS. Also, OOD (orange) and in-distribution (blue) error are matched, showing OOD generalization.

## 5 Future work

DynaDojo is still a work in progress. As of our most recent release, our platform has certain limitations worth noting: While we implement parallel computing across all `Challenges`, the `FixedError` challenge is still especially resource-intensive and susceptible to noise. We aim to implement more sophisticated root-finding search algorithms to replace our exponential search.

To improve the rigor of our scaling results, we would like to move to a more objective measure of system complexity (e.g., the intrinsic dimension of the objective landscape [23]). We also seek to develop scaling metrics that summarize the results plotted by DynaDojo. Furthermore, we would like to define a unified generalization metric that captures an algorithm's capacity to work on OOD test data, across scales of complexity, and on different dynamical systems altogether.

We will, of course, always look for ways to include more state-of-the-art algorithms and dynamical systems of interest. In particular, we would like to develop simple wrappers for OpenAI Gym environments and algorithms to immediately accommodate their vast library, and vice-versa, wrapping DynaDojo `Systems` and `Algorithms` for OpenAI Gym.

To further broaden the scope and applicability of DynaDojo, we plan on introducing new `Challenges` of interest to the community. For example, we aim to incorporate an optimal control challenge involving stabilizing a system around a target trajectory, a transfer learning challenge focusing on fine-tuning to new system data, a multi-task learning challenge around maintaining consistent performance across multiple dynamical systems, and a curriculum learning challenge focusing on leveling up system complexity without retraining an algorithm from scratch. All of the `Challenges` can also be extended to measure prediction/control error on multiple timescales.

Lastly, in light of the emerging importance of scaling laws in deep learning, we hope to incorporate new scaling dimensions. These will include the number of model parameters, computational cost of training, and activation sparsity, on top of the three existing scaling dimensions. By including these dimensions, we aim to offer a more comprehensive view of how algorithms scale.

## Author contributions

Using the Contributor Roles Taxonomy (CRediT)[2]:

- LMB contributed to the software (lead on design, documentation; support on algorithms/systems), data curation (equal on demo notebooks).
- TB contributed to the visualization (lead on creative direction; co-lead on figures), software (lead on algorithms/systems; support on documentation), data curation (equal on demo notebooks), writing (editing).
- CT contributed to the writing (co-lead), software (lead on reproducibility, parallelization, testing; support on design), visualization (co-lead on figures), and led the investigation, validation.
- RM contributed to the software (support on algorithms/systems).
- MK led the conceptualization, supervision, project administration, methodology, and contributed to the writing (co-lead), software (all-around support).

## Acknowledgments

LMB was supported by Stanford's CURIS program. TB was supported by Stanford's Bioengineering REU program. MK was supported by an NSF Graduate Research Fellowship. The authors thank Stanford's Brains in Silicon Lab, Northwestern's Delta Lab, and Sean Yoon for helpful feedback on the writing. This work used Stanford's Sherlock and Northwestern's Quest clusters to collect results.

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
