# OpenReview forum: "DynaDojo: An Extensible Benchmarking Platform for Scalable Dynamical System Identification"
_NeurIPS.cc/2023/Track/Datasets_and_Benchmarks — NeurIPS 2023 Datasets and Benchmarks Poster_

### Official Review · Reviewer_rZxW · 2023-07-19
**The paper introduces a platform for a limited problem in system identification of dynamical systems that requires significant extensions.**

**Rating:** 3
**Confidence:** 4
**Clarity:** The presentation could be improved. D…

**Strengths:**

1. The paper presents an extensible framework to study a very important and relevant problem of system identification, which has several applications across domains.

2. The paper is well-organized and the explanations highlight well the features of DynaDojo that allow exploration of active learning algorithms to improve sample efficiency. This expands the research landscape in dynamical system identification by providing access to the data-generating process.

3. The benchmarks, systems and models implemented provide a reasonable initial set of problems that demonstrate the benefits of the approach.



**Additional Feedback:**

1.	The paper presents the problem using control dynamical systems that include a control signal u(t). The control signal plays a very significant role in system identification since it is desirable to excite adequately the system dynamics. How is the control signal selected in the platform?

2.	Another aspect that is critical is the observation map and noise that can affect system identification methods. It seems that it will not be difficult to introduce this capability in the platform.

3.	The time horizon of the trajectories and the selection of the sampling frequency also play an important role. It seems that the platform currently focuses only on initial conditions and needs to be extended considerably to impact dynamical system identification benchmarks.

4.	Can you be more specific and provide some concrete examples of measures of system’s complexity for the dynamical systems supported?

5.	Is there a way to somehow quantify/measure/approximate the trade-off between sample efficiency and generalizability of the model using the framework? This would be interesting as these phenomena can be conflicting at times.

6.	A typical task in system identification is to explore different models and choose the optimal one including structure and parameters. Optimality can be defined with respect to generalization/sample efficiency/system complexity but also in terms of the underlying objective. For example, control tasks such as stabilization or tracking may have different requirements about the identified model. State estimation and forecasting can be additional objectives. Perhaps, an example demonstrating such a task will be useful to highlight the proposed platform.

7.	Another aspect of system identification is that it includes an optimal design of experiments to generate the data required. This goes beyond active learning for sampling new initial conditions and may include exogenous inputs both controlled and disturbances from the environment and other factors. In summary, the proposed platform is preliminary focusing on a limited problem.

8.	The presentation of the paper could be improved. Examples include:
a.	Stock market forecasting and human social interactions are mentioned in the introduction, but it is not clear how they are related to the technical results in the paper.
b.	It seems that the paper refers to continuous-time and discrete-time dynamical systems (and not continuous and discrete which typically refer to the domain of the state in system theory).
c.	The learning problem presented in Section 3 that aims to predict future states from state trajectories is more limited than system identification.
d.	It is not clear what is the definition of “reliability of generated trajectories?”

9.	It is not clear if Figure 3 and Figure 8 are derived from real data and what these data are or they are just used to illustrate what it is possible using the platform.


**Correctness:**

While the claims in the paper are mostly correct, the problem is very limited and many details (e.g., system complexity) are missing.

**Documentation:**

A URL with sufficient details is provided.

**Ethics:**

No concerns

**Limitations:**

Although the paper mentions some limitations, system identification is much more complex than presented in the paper.

**Opportunities For Improvement:**

1.	Although the initial platform is reasonable, there are multiple aspects of system identification beyond selecting initial conditions that are not adequately addressed including selection of control inputs, selection of sampling frequency for continuous dynamical systems, signal-to-noise ratio of the observations, selection of time horizon for the trajectories, model structure, algorithms for system identification, and so on. More details can be found in the comments below.

2. The explanation related to the different type of “generalizations” could benefit from technical details and evaluation. How is the platform used to evaluate these types of generalization?

3. The paper mentions existing benchmarks in dynamical system identification, such as static datasets, however, it does not discuss or compare with such approaches. Although the ability to perform active learning is important, how the proposed platform compare with other capabilities of existing benchmarks and toolboxes. Is it possible and is there a plan to start adding capabilities (e.g., selection of control inputs etc.)?

4. System identification has been a very active research area with a multitude of models and methods for various dynamical systems. The proposed platform is rather limited. For example, it supports only four classes of dynamical systems that are represented by well-defined equations, which seem to allow choosing some type of complexity. How difficult is to apply similar methods more generally to dynamical systems that can be simulated without explicit mathematical equations?


**Relation To Prior Work:**

System identification is a very active area with many different aspects. Just focusing on sample efficiency for a simple set of problems is not adequate.

**Summary And Contributions:**

The paper introduces a benchmark platform designed for the system identification problem for complex dynamical systems. The platform, DynaDojo, measures the sample efficiency and generalization of algorithms to out-of-distribution data, different levels of complexity for various classes of dynamical systems. The platform also provides a user-friendly API that supports various models, user-defined dynamical systems, and evaluation metrics. Computational efficiency is prioritized through resource-efficient training and parallel processing strategies. The paper evaluates multiple baseline algorithms using DynaDojo across four classes of dynamical systems and aims to establish it as a unifying benchmarking platform for system identification.

---

> ### Author Response · Authors · 2023-08-14
> **Clarifying System Parameters and Suggested Capabilities**
>
> Dear Reviewer,
>
> Thank you for taking the time to provide such detailed and sincere feedback. Your insights have highlighted areas where our paper could benefit from clearer explanations. In particular, we want to emphasize that many of the key system parameters you mentioned are indeed incorporated in our platform. We plan to improve how we communicate these details in our revision, likely with expanded explanations in the Challenge Customizability paragraph (line 217), Appendix A, and on GitHub.
>
> To address your concerns:
> - Time Horizon & Sampling Frequency: When instantiating a Challenge in DynaDojo, users can specify the number of timesteps in a System’s trajectory. This effectively sets the sampling frequency for continuous-time systems. Developers adding new dynamical Systems determine the absolute time interval relevant for simulating trajectories.
>
> - Control: Beyond just the control/selection of initial conditions, our platform executes controls received from a user’s model over a user-defined control horizon. This allows for the type of optimal experimental design you described. Additionally, we provide the capability to enforce power constraints on the control inputs.
>
> - Signal-to-Noise Ratio: Users have the option to introduce noisy dynamics through a boolean flag. Expert users can further customize the default noise level.
>
> - System Complexity: Developers adding a new dynamical System to the platform determine a metric for that System’s complexity. This could be the rank of the dynamics matrix or another metric deemed appropriate by the developer.
>
> Our overarching aim is to offer a balance: We want to expose critical system parameters without overwhelming users. For those desiring more precision, nearly all our system parameters are adjustable.
>
> There were also a few points from your review we wanted to better understand:
> 1. Could you elaborate on what you mean by "model structure" and "algorithms for system identification"? Our primary contribution is the benchmarking platform, and we remain agnostic to the specifics of a user’s learning algorithm implementation.
> 2. We'd appreciate more insight into "dynamical systems that can be simulated without explicit mathematical equations". While we're not immediately aware of such systems, they would, in principle, be compatible with our platform.
> 3. You suggested an example demonstrating a task with multiple objectives (e.g., control, stabilization, tracking). Given our focus on introducing new generalization metrics in this release, we centered on forecasting as the primary objective. We plan to add more control objectives in the next major release and would value your guidance on balancing the scope of this paper with the inclusion of more objectives.
>
> We are committed to making the necessary revisions to enhance the clarity and depth of our paper, and we sincerely value your continued guidance in this process. We hope that our revisions will address the concerns and elevate the contribution of our work to the field. Thank you once again for your constructive feedback.

---

> > ### Comment · Reviewer_rZxW · 2023-08-23
> >
> > Thank you for providing additional details about the capabilities of the platform. Being able to support key system parameters will be critical for the platform.
> >
> > In terms of the questions:
> >
> > 1. For linear models, for example, there are various possibilities for structure - state-space, polynomial forms, etc. Even if someone considers only linear state-space models, model structure selection includes selecting the model order and input delay. Selecting the structure affects model quality metrics that are dependent on the loss function, regularization, etc. Although many of the details can be found in the system identification literature a good source is the documentation of the Matlab system identification toolbox that is used extensively (the documentation is openly available.)
> >
> > 2. Many modern simulators of complex systems allow modeling systems as interconnection of components that may be described by equations but also by imperative code. In this case, there are no explicit equations of the overall system. There are also black-box simulators (e.g., flight and car simulators.) I understand that this is out of the scope of the paper but it is closer to identifying a model for experimental systems using system identification.
> >
> > 3. Comparing the identified model output with the data can be the primary objective. But this needs to be performed for different control input data.

---

### Official Review · Reviewer_Nmnd · 2023-07-21
**Review of DynaDojo, a benchmark of dynamical systems solvers**

**Rating:** 7
**Confidence:** 4

**Strengths:**

Analyzing the use of data driven methods is considerably more opaque than traditional methods. Often, a model builder will publish the results of a large model, trained with unknown amounts of data for unknown amounts of time, making it look very impressive. I believe this platform is very useful to the scientific community that is thinking about using such methods, because it aligns with things that researchers care about. For example, how do methods compare if one cares about achieving a fixed accuracy on a problem. Or, given that one has a fixed amount of data available, which method performs better or has a lower uncertainty for a certain regime.
Additionally, the the API seems well architectured and general enough to allow for broader comparisons along the chosen cross sections of analysis.


**Additional Feedback:**

N/A

**Clarity:**

The setup of the benchmark seems very clear and the explanations are clearly written, especially Figure 3 which gives the reader a general idea of what we expect from the results. However, I believe Section 6 needs to be made clearer (see above Opportunities for Improvement).


**Correctness:**

The evaluation methods for this benchmark are excellent for targeting real world use cases of optimizing model type, training data complexity and system complexity.

**Documentation:**

The API seems very clean and easy to implement.


**Limitations:**

It seems that the benchmark currently does not take a look at the computational complexity of the methods being considered, something which is (arguably) as important as the other analyses in Figure 3. I understand that this is hard to do from an engineering point of view, either by asking derived classes of AbstractModel to self report parameter sizes or somehow track them like PyTorch does, but I believe it deserves a discussion.
Additionally, the generality of the API means that it is difficult to use for more specialized applications. For example, implementing a mesh based system is possible but you’d have to throw away mesh specific information. While I don’t expect this platform to handle every specialized case, I do think that it deserves discussion especially as other related work can in some cases.


**Opportunities For Improvement:**

Figure 6 is a showcase of what can be shown using the benchmarking system, but it deserves more discussion and a better presentation stylistically. Firstly, I believe more discussion about the different graphs is warranted to convey the significance of the findings. For example, the first graph of 6a shows that the minimum error that can be achieved is around 1e-6, 6c sees a linear relationship past a certain point between system complexity and training samples (non trivial). More than just showing what happens with these models, I think that showing what can be ascertained through the whole organization of the benchmark is needed. Secondly, the style needs to be cleaned up, as there are two legends in 6b, varying usage of error bars and transparent spread, different sharpnesses of the images, etc.
Another thing that deserves mention is that MSE is not always the most important thing when analyzing dynamical systems. Sometimes there exist more desirable properties, such as energy conservation or dissipation, relative metrics that take into account the magnitude of the solution particularly in OOD settings, etc.


**Relation To Prior Work:**

I think the discussion is generally broad enough, however I think that more comparison needs to be done with “Benchmark Suites”. I don’t think that the distinction “supporting the addition of any kind of dynamical system” really holds, because what you describe in the Problem Setup in terms of dynamical systems can also be benchmarked with existing PDE benchmarking tools because it is doing explicit time stepping on a PDE.


**Summary And Contributions:**

This paper presents a benchmark of dynamical systems solvers, with three general tiers of extensibility:
* Models that are to be evaluated, such as Neural ODE.
* Systems to evaluate on, such as cellular automata.
* Challenges, or slices across tunable parameters of the above two. For example, for a fixed solution tolerance, what does the graph of training examples necessary to reach that tolerance look like for varying system size.

---

> ### Author Response · Authors · 2023-08-24
> **Discussion of Other Metrics, Computation, and Specialized Systems**
>
> Dear Reviewer,
>
> Thank you very much for your discussion of how our work contributes to more transparently understanding model performance as well as for your feedback. To address your concerns:
>
> - **Other metrics**: We agree with you that other metrics besides MSE are important to measure and those can be achieved by updating the `calc_error` function of each System. Custom error metrics like the ones you mentioned are able to be defined by the user for each System. For example, we use a modified version of Hamming distance to measure error in the `CASystem`.
>
> - **Computational complexity**: Currently, to test models with various computational complexities, a user can develop multiple versions of the model and then run each one in tandem on a Challenge. We agree though that being able to more seamlessly explore how a model performs by automatically iterating computation is important, and would look to implement this directly into our Challenges in further versions of our platform.
>
> - **Specialized systems**: DynaDojo is unopinionated about how data is generated. If the data can be packaged as a numpy matrix of trajectories, the platform can use it. As such, the API can handle PDEs, meshes, MDPs, and more. We are open to exploring more of these specialized systems.
>
> - **Results**: We are continuing to work on addressing the visual changes mentioned for Figure 6 as well as improving the discussion of tangible learnings one can ascertain from a model, thank you for your feedback on this area.
>
> Thank you so much for your feedback and effort reviewing our submission!

---

> > ### Comment · Reviewer_Nmnd · 2023-08-30
> >
> > Thanks for the comments. The addition of the new systems was good to see and confirms my original rating.

---

### Official Review · Reviewer_j4gM · 2023-07-24
**Interesting approach with some issues**

**Rating:** 6
**Confidence:** 3
**Correctness:** Evaluation and experiment design is a…

**Strengths:**

The main strength of the paper is unifying a number of existing test environments in a single framework. Identifying common feature of existing environment, in order to unify them, is the main contribution.
Four prototypical classes of dynamical systems are used to illustrate the framework.

**Additional Feedback:**

1) To use the framework, one needs to spend some time to understand how these classes work together, even if the API is "*user-friendly*" (line 286). Figure 4 illustrates this software engineering complexity, as well as the below quotes from the supplementary material:
- line 403 - 408 (supp. material): "The API is structured around three core classes: AbstractSystem,
 AbstractModel, and Challehnge. Their relationship is shown in the UML diagram shown in Figure 5 and described in detail below.
 In DynaDojo’s evaluation pipeline, a Challenge instance calls evaluate on a subclass of AbstractModel. Instances of the concrete
 subclass are repeatedly initialized and fit on data generated from a instance of an AbstractModel subclass."
[...]
line 434 - 438 (supp. material): "*In DynaDojo, models must subclass from AbstractModel and implement several abstract methods: fit, predict, and (if the model uses control) act. fit trains a model on a collection of trajectories in embedding space. predict predicts the evolution of a set of initial conditions over a given number of timesteps. The first element of the trajectory should always be the initial condition. act allows models to set a control for a given control horizon (described below).*"

This is not a critique though, just an observation (since the word "*user-friendly*" appears multiple times, lines 78, 225, 286).

2) A large literature exists on PDEs as well, in addition to ODEs. Would your framework be able to incorporate these as well?

3) line 264-266: "*Presumably [sic!], the model is not approximating the true dynamics but rather overfitting to the particular evolutions of the training initial condition. This approximation breaks down when the orthant of the state space that the initial conditions belong to changes at test time.*"
Couldn't you simply compare with the explicit solution of the model in order to know with certainty whether overfitting occurred?

---

I would recommend to the authors, based on my entire review, to address all the issues I mentioned and polish the paper some more, and, in particular, provide a more compelling Jupyter notebook to test the framework and to offer more examples of dynamical systems a double-digit number would be good), in particular, those where earlier papers (e.g., the SINDy paper I referenced in a section above) already provided some results, so that we can compare against earlier approaches. Based on this, I can only award it a score of **5**.

If that could be achieved, I think it would be a good submission. Currently, it seems more like a workshop submission, due to the small number of dynamical systems; I would have accepted it as a workshop submissions.

**Clarity:**

line 121: "Such continuous systems can be recast as discrete systems" What do you precisely mean by that?

line 273: "*Primarily, while training is conducted on controlled systems, these controls are not incorporated into testing. Ideally, it would be beneficial to independently evaluate the generalization and sample efficiency of control outside of system identification*" I did not understand this fully, could you illustrate it with a concrete example?

- typos:
line 183: "how error changes" -> "how the error changes"
line 408 (supplemental material): "a instance" -> "an instance"
line 463 (supplementary material): "for the the" -> "for the"
line 588 (supplementary material): "Generalizaion" -> "Generalization"

**Documentation:**

Hosting and licensing does not seem to have been mentioned. Reproducing the plots regarding dynamical system identification should be possible since a `playground.ipynb` file is on GitHub. Although currently it is saved with errors, and also rather small (see a screenshot of the current state at https://ibb.co/khhJWNb).

**Limitations:**

It would seem to me that the only limitations to their method are inherent to the ones present in the `sklearn` library (if that is the case, it would be good if the authors could briefly mention this in the paper as well).

**Opportunities For Improvement:**

While it is important to have some example that illustrate the framework that was introduced, four classes of dynamical systems are too small. One of the benefits of environments like OpenAI's gym is that a large number of environments exists to test on. The authors state that their codebase can easily be extended with further examples, but I am wondering why they haven't extended them.

The issue mentioned above is the main problem at the moment. A number of smaller issues are mentioned in the sections on *Limitations*, *Clarity*, *Relation to Prior Work*, *Documentation* and *Additional Feedback*.

**Relation To Prior Work:**

Previous contributions are discussed, but important references seem to be missing, such as the SINDy paper [1]. It would be good to incorporate the results from there into DynaDojo, to see if the results match (if they do, it would increase my confidence that the implementation of DynaDojo is indeed correct and consistent with existing literature).

[1] SL Brunton et al., Discovering governing equations from data by sparse identification of nonlinear dynamical systems

**Summary And Contributions:**

An environment for working with dynamical systems is presented. The environment is structured along three dimensions: "Challenges" (what you want to test, such as how the error changes when you add more samples), "Systems" (the dynamical system class you want to test" and "Models" (the neural network you want to use to approximate the model).
Thus, the authors unify existing approaches, in cases where the investigation could be split along these dimensions. Since few dynamical systems are offered at time of submission, Figure 4 is the essential figure that indicates how to extend the environment, given the framework they provided (the supplementary material contains the software engineering counterpart to Figure 4, where it is explain how and which Python classes that the authors offer need to be extended).

---

> ### Author Response · Authors · 2023-08-24
> **System Updates, User-Friendliness, and More**
>
> Dear Reviewer,
>
> Thank you very much for your feedback and your discussion of our work’s contributions. We wanted to share that we have fixed the typos, and updated the mentioned sentences for better clarity (please let us know if we can make any further improvements). Thank you for your specific mention of the SINDy library; we have implemented the SINDy models and recreated some preliminary results in our demo. We have added a note in the Results section about the model overfitting seen in the figure by comparing it with the explicit solution as you mentioned and further confirmed this with visual analysis of the limiting behavior. With the addition of our 13 Systems, we are now at 17 systems, and have provided a demo for each of them as we agree that this makes the platform’s contributions more compelling.
>
> With the demo and Figure updates as well as tailored subclasses for related groups of Systems, we hope to more clearly show the seamless interactions of the parts of our platform. Our goal is that these updates will provide clearer understandings of the platform for news, and we are open to any other suggestions on how to further reduce the complexity of understanding our system. We have also updated the GitHub with licensing information.
>
> We wanted to ask if there were specific limitations of sklearn you were considering that we should mention. In addition, our platform can handle PDEs, and we will explore implementing them.
>
> Thank you for your time reviewing our platform!

---

> > ### Comment · Reviewer_j4gM · 2023-08-31
> >
> > Thank you for the response, the paper reads much better now.
> >
> > line 259-261: "We have recreated findings from Brunton et al about the efficacy of various differentiation methods with SINDy on Lorenz system, available in demonstrations on the API’s GitHub. We include numerous more sample results in Appendix D.".
> >
> > I think on Github, the correct path is rather https://github.com/FlyingWorkshop/dynadojo/blob/main/demos/sindy_demo1.ipynb.
> > Well done nonetheless on recreating the system from the SINDy paper. It would be great though, to include some of these results in paper as well.
> >
> > In a way, the GitHub now seems to be better than the paper, since it contains many nice Jupyter notebooks. Strangely, the authors haven't included some of these plots in the supplementary material.
> >
> > I'm still doubtful about the handling of PDEs, since it seems no example on PDEs are given. The authors claimed that their platform would be able to incorporate PDEs, but PDEs are much harder to work with than ODEs; so while conceptually I can see the authors' viewpoint that PDEs could be supported, the authors might find significant hurdles in actual implementation (I'd be very curious about what the authors find). **I'd need more on the PDE front in order to unequivocally increase my score; because I can't give 0.5 increments in score increases, to a 5.5, I'm increasing to 6**.  Because there is little time left, I'd rather recommend the authors to drop the PDE mentioning entirely, rather than mentioning it, but providing no counterexamples, since that confuses the reader.

---

### Official Review · Reviewer_33kq · 2023-07-31
**A Benchmark for Dynamical Systems**

**Rating:** 6
**Confidence:** 2
**Correctness:** To the best of my knowledge the claim…

**Strengths:**

The benchmark is well-motivated, and feels like it fills a gap in the existing work.  From a usability perspective, this benchmark seems much more user friendly than many of the other submissions that I have reviewed for this track at NeurIPS.

**Additional Feedback:**

None.

**Clarity:**

The work is well-motivated for a general audience (it could be more compact in places for an ML audience).

**Documentation:**

The documentation on the GitHub link is sparse, but the submitted supplement seems thorough enough.

**Ethics:**

No concerns.

**Limitations:**

Yes, the authors discussed the limitations of the approach.

**Opportunities For Improvement:**

For me what made Open AI Gym so successful was the collection of core challenge problems that could get experts and non-experts interested in RL.  It would be nice to see similar application driven examples in this benchmark (I don't know if you average person would get super excited about cellular automata or generic linear dynamical systems).  Of course, including state-of-the-art algorithms available for comparison would also encourage people to use the system.

**Relation To Prior Work:**

I believe so. However, this is not my area of expertise, so I don't know if there are existing benchmarks that overlap the proposed one that have been omitted.

**Summary And Contributions:**

The authors propose a generic and extensible benchmarking system for dynamical systems whose core design is inspired by OpenAI Gym.  The benchmark is designed to assess generalization (both within the same problem class and outside) and sample complexity as well as to provide an evaluation framework at varying levels of dynamical system complexity.

---

> ### Author Response · Authors · 2023-08-24
> **Real-World Systems, and Clarity Updates**
>
> Dear Reviewer,
>
> Thank you for your comments on the user-friendly nature on the platform and on its contribution. We agree with your feedback that DynaDojo is more successful with more real-world Systems. We specifically prioritized adding Systems focused on human social interactions, biochemical processes, and ecological interactions. We have also thoroughly updated the GitHub documentation, and would be happy to receive any additional feedback on these updates.
>
> We wanted to ask if there were specific sections/sentences that you had in mind for where the paper writing could be more compact for an ML audience.
>
> Thank you very much!

---

### Author Response · Authors · 2023-08-24
**Revisions for Major Themes Across Reviews**

Dear Reviewers,

Thank you all for your writing such detailed comments and helpful suggestions for this work. We are encouraged by the value you see in this platform, and are excited to share updates that address the major themes across your reviews:

- **We have added 13 new systems, many of which are real-world application-driven**: We have added *Lotka-Volterra* systems for simulating foodwebs; *Bounded-Confidence Opinion* systems to measure bias and polarization; a *Kuramoto* systems to measure biochemical synchronization processes like heart pacemaker cells and fireflies, as well as *Epidemic* systems, an *N-Body* system, and a generalized *Lorenz* system. More details are available in Figure 2 and Table 3. We have also updated the code with `AbstractSystem` subclasses of `SimpleSystem`, `EpidemicSystem`, and `OpinionSystem` to make it easier to quickly add new systems that are related to existing systems.

- **Updated demos and documentation**: On GitHub, you can find demos for each of the 17 Systems and the 3 Challenges to start easily analyzing models. We have updated the GitHub documentation as well. Lastly, we have updated the paper Figures 1, 3, and 9  with labels of each generalization and sample efficiency metric is measured, and added a new Figure 5 to explain all the adjustable parameters in DynaDojo.

- **Tie-ins with prior work**: DynaDojo can robustly support existing dynamical systems generators and models, including PDEs. We have specifically incorporated models from Rossetti et al’s NDlib library as well as models from Brunton et al’s SINDy platform, and these are available in our demos. We have also added new Table 1 that compares DynaDojo with related existing benchmarks.

We have highlighted our changes so far in red text color (please see new PDFs). In the remaining time of the review period, we plan to add details of our newly added Models/Systems to the Appendix and to further address your comments regarding the Results and Limitations section. Please see the specific comments under your reviews for more details and discussion.

Thank you all for your time and effort reviewing DynaDojo!

---

### Author Response · Authors · 2023-08-31
**Final Updates: PDE Systems, Visuals in Paper**

Dear Reviewers,

Thank you all again for your continued feedback and effort in reviewing DynaDojo. We wanted to write to share two final updates:

- **We have added 3 new PDE systems, bringing DynaDojo to a total of 20 systems**: These new systems include a 2D Heat Equation System that we simulate on a spatiotemporal grid, whose size we iteratively increase to make more complex. They also include a Black-Scholes-Barenblatt system (used heavily for financial modeling) and a Hamilton-Jacobi-Bellman (often used for optimal control) that are solved using forward-backward stochastic differential equations so that we can simulate dimensions that are significantly higher than typical numerical solvers for PDEs (e.g., dim=10,50,100). We have provided a FBSNN subclass of AbstractSystem to make it easier to continue to add PDEs in higher-dimensions. We recognize your continued feedback in integrating this important field of dynamical systems into our project and are excited to share the addition of these PDE systems.

- **Thorough Appendix with visual examples**: We have added technical details and visual graphs of each of our System classes and Models to the Appendix. In our GitHub demos, it is possible to simulate a diverse range of Systems, dynamics parameters, initial_conditions, noise, and more, and we chose to bring in the most representative and helpful examples directly into the paper.

Our changes continue to be highlighted in red text color. We have also addressed some of the comments made in regards to the learnings of the Results section, as well as cleaning up the graphs in that section.

Thank you very much all for your time and effort reviewing DynaDojo!

---

### Decision · Program_Chairs · 2023-09-22

**Decision:**

Accept (Poster)

**Comment:**

The paper presents a solid benchmark for dynamical systems, and 3 out of 4 reviewers were in favor of the contribution being accepted. The issues raised by reviewer rZxW have (to a substantial degree) been addressed in the updated version of the paper, and I hence recommend acceptance.